# What does mitogenomics tell us about the evolutionary history of the *Drosophila buzzatii* cluster (*repleta* group)?

**Nicolás Nahuel Moreyra**[1,2]*, **Julián Mensch**[1,2], **Juan Hurtado**[1,2], **Francisca Almeida**[1,2], **Cecilia Laprida**[1,3], **Esteban Hasson**[1,2]*

**1** Departamento de Ecología, Genética y Evolución, Facultad de Ciencias Exactas y Naturales, Universidad de Buenos Aires, Ciudad Autónoma de Buenos Aires, Argentina, **2** Instituto de Ecología, Genética y Evolución de Buenos Aires, Consejo Nacional de Investigaciones Científicas y Técnicas, Ciudad Autónoma de Buenos Aires, Argentina, **3** Instituto de Estudios Andinos, CONICET/UBA, Ciudad Autónoma de Buenos Aires, Argentina

* nmoreyra@ege.fcen.uba.ar (NNM); ehasson@ege.fcen.uba.ar (EH)

**Data Availability Statement:** Mitogenomes and their corresponding annotations are deposited in GenBank under accession numbers MN551230 (*D. antonietae*), MN551231 (*D. borborema*),

## Abstract

The *Drosophila repleta* group is an array of more than 100 species endemic to the "New World", many of which are cactophilic. The ability to utilize decaying cactus tissues as breeding and feeding sites is a key aspect that allowed the successful diversification of the *repleta* group in American deserts and arid lands. Within this group, the *Drosophila buzzatii* cluster is a South American clade of seven closely related species in different stages of divergence, making them a valuable model system for evolutionary research. Substantial effort has been devoted to elucidating the phylogenetic relationships among members of the *D. buzzatii* cluster, including molecular phylogenetic studies that have generated ambiguous results where different tree topologies have resulted dependent on the kinds of molecular marker used. Even though mitochondrial DNA regions have become useful markers in evolutionary biology and population genetics, none of the more than twenty *Drosophila* mitogenomes assembled so far includes this cluster. Here, we report the assembly of six complete mitogenomes of five species: *D. antonietae*, *D. borborema*, *D. buzzatii*, two strains of *D. koepferae* and *D. seriema*, with the aim of revisiting phylogenetic relationships and divergence times by means of mitogenomic analyses. Our recovered topology using complete mitogenomes supports the hypothesis of monophyly of the *D. buzzatii* cluster and shows two main clades, one including *D. buzzatii* and *D. koepferae* (both strains), and the other containing the remaining species. These results are in agreement with previous reports based on a few mitochondrial and/or nuclear genes, but conflict with the results of a recent large-scale nuclear phylogeny, indicating that nuclear and mitochondrial genomes depict different evolutionary histories.

## Introduction

Almost all mitochondrial genomes, the "mitogenome", can be assembled directly from genome or even transcriptome sequencing datasets [1, 2]. The exponential development of

MN551232 (*D. buzzatii*), MN551233 (*D. koepferae B*), MN551234 (*D. koepferae A*), and MN551235 (*D. seriema*).

**Funding:** This work was supported by University of Buenos Aires (UBA), CONICET and ANPCyT grants awarded to EH. UBA: http://www.uba.ar/; CONICET: https://www.conicet.gov.ar/; ANPCyT: https://www.argentina.gob.ar/ciencia/agencia. The funders had no role in study design, data collection and analysis, decision to publish, or preparation of the manuscript.

**Competing interests:** The authors have declared that no competing interests exist.

next-generation sequencing (NGS) technologies, together with efficient bioinformatic tools for the analysis of genomic information, has allowed efficient assembly of mitochondrial genomes, giving rise to the emergence of the mitogenomics era [3]. Mitogenomics has been very useful in illuminating phylogenetic relationships at various depths of the Tree of Life, e.g., among early branching of metazoan phyla [4], among crocodilians and their survival at the Cretaceous-Tertiary boundary [5], primates [6], the largest clade of freshwater actynopterigian fishes [7] and Anura, the largest living Amphibian group [8]. Also, mitogenomic approaches have been used to investigate evolutionary relationships in groups of closely related species (e.g. [9]). In animals, the mitochondrial genome has been a popular choice in phylogenetic and phylogeographic studies because of its mode of inheritance, rapid evolution and the fact that it does not recombine [10]. Such physical linkage implies that all regions of mitogenomes are expected to produce the same phylogeny. However, the use of different mitogenome regions or even the complete mitogenome may lead to incongruent results [11], suggesting that mitogenomics sometimes may not reflect the true species history but rather the mitochondrial history [12–16]. Moreover, there is evidence suggesting that mtDNA genes are not strictly neutral markers, casting doubts on its use to infer the past history of taxa [17]. Inconsistencies across markers may result from inaccurate reconstructions or from actual differences between genes and species trees. In fact, most methods do not take into consideration that different genomic regions may have different evolutionary histories, mainly due to the occurrence of incomplete lineage sorting and introgressive hybridization [18–20].

Over the last century, the *Drosophila* genus has been extensively studied because of the well-known advantages that several species offer as experimental models. A remarkable feature of this genus that comprises more than two thousand species [21] is its diverse ecology: some species use fruits as breeding sites, others use flowers, mushrooms, sap fluxes, and fermenting cacti (reviewed in [22–25]). The adoption of decaying cacti as breeding sites occurred more than once in the evolutionary history of *Drosophilidae* [26, 27] and is considered a key innovation in the diversification and invasion of American deserts and arid lands by species of the *Drosophila repleta* group (hereafter the *repleta* group) [26]. Many species in this group are capable of developing in necrotic cactus tissues and feeding on cactophilic yeasts associated to the decaying process [28–35].

The *repleta* group comprises more than one hundred species [23, 36–39], however, only one of the more than twenty complete (or nearly complete) *Drosophila* mitogenomes assembled so far belongs to a species in this group (checked in GenBank, March 28, 2019), *D. mojavensis* (GenBank: BK006339.1). The latter, the first cactophilic fly to have a sequenced nuclear genome [40], is a member of the *D. mulleri* complex, an assemblage of species that belongs to the *D. mulleri* subgroup, one of the six species subgroups of the *repleta* group [37].

The *D. buzzatii* complex is the sister group of the *D. mulleri* complex [26]. It diversified in the Caribbean islands and South America, giving rise to the *D. buzzatii* cluster (hereafter the *buzzatii* cluster), and the *D. martensis* and *D. stalkeri* clusters [41]. The former is an ensemble of seven closely related species including *D. antonietae* [42], *D. borborema* [43], *D. buzzatii* [44], *D. gouveai* [42], *D. koepferae* [45], *D. serido* [43], and *D. seriema* [46]. All species are endemic to South America (Fig 1), except the semi-cosmopolitan *D. buzzatii* that reached a wide distribution following human mediated dispersal of prickly pears in the genus *Opuntia* (Caryophillales, Cactaceae) in historical times [35, 47, 48]. These species inhabit open areas of sub-Amazonian semidesert and desert regions of South America, where flies use necrotic cactus tissues as obligatory feeding and breeding resources [35, 49]. Regarding host plant use, *D. buzzatii* is an *Opuntia* specialist [31], considered an ancestral condition [26]. However, *D. buzzatii* has also been reared from necrotic columnar cacti [35]. The remaining species are mainly columnar dwellers although *D. antonietae* and *D. serido* can also use O. monacantha [49],

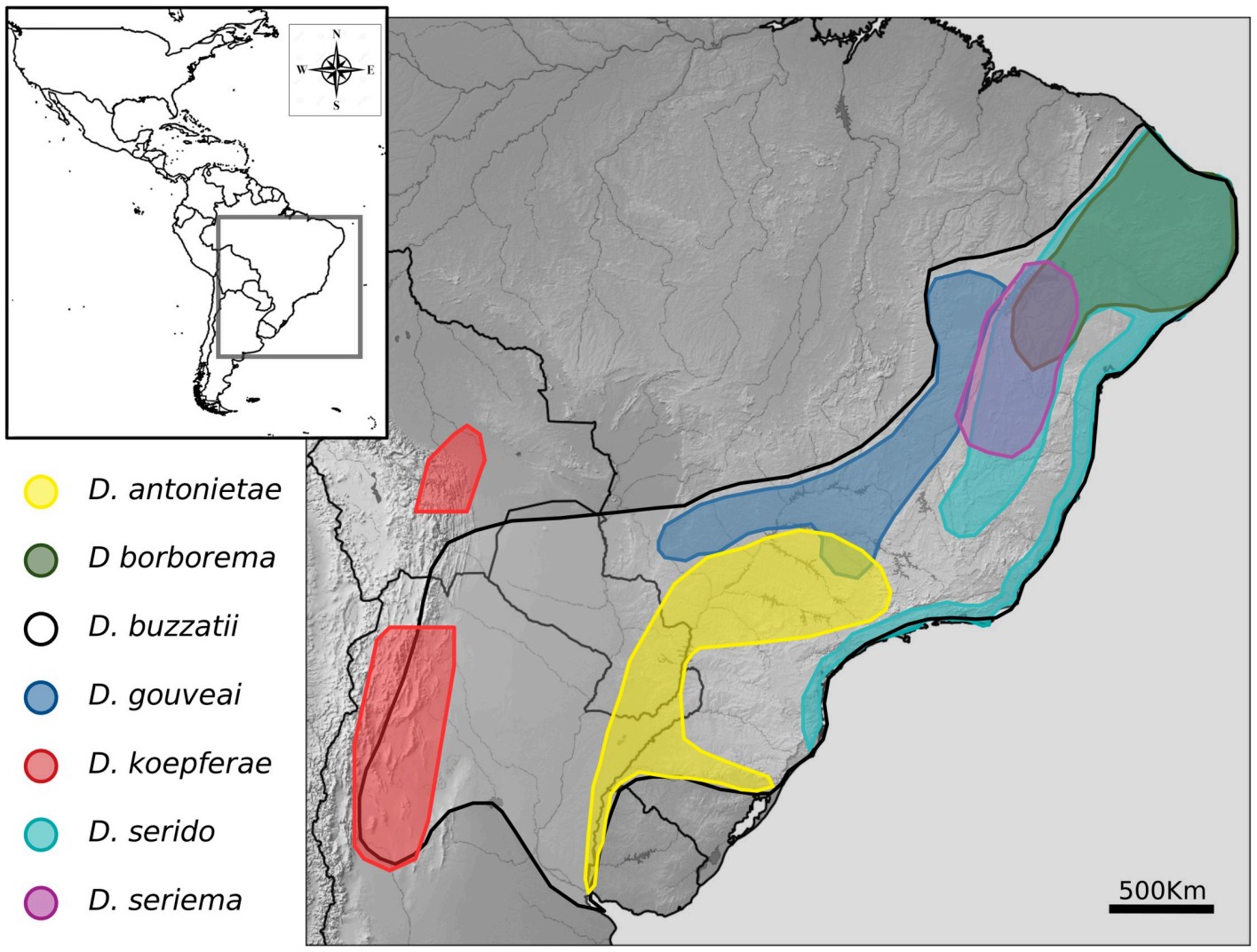

**Fig 1. Geographical distribution of *buzzatii* cluster species modified from reference [50].**

while *D. koepferae* has also been recovered from decaying cladodes of *O. monacantha*, *O. quimilo* and *O. sulphurea* [31].

Species of the *buzzatii* cluster are almost indistinguishable in external morphology, however, differences in the morphology of the male intromittent organ (aedeagus) and polytene chromosome inversions provide clues to species identification (reviewed in [35, 48, 51]). The cluster has been divided into two groups based on aedeagus morphology, the first includes *D. buzzatii* and the remaining species compose the so-called *Drosophila serido* sibling set -*serido* sibling set from hereafter- [48]. In turn, analysis of polytene chromosomes revealed four informative paracentric inversions that define four main lineages: inversion $5g$ fixed in *D. buzzatii*, $2j^9$ in *D. koepferae*, $2x^7$ shared by *D. antonietae* and *D. serido*, and $2e^8$ shared by *D. borborema*, *D. gouveai*, and *D. seriema* [41, 52]. However, neither genital morphology nor chromosomal inversions are useful for inferring basal relationships within the cluster.

Pre-genomic phylogenetic studies based on a few molecular markers generated debate since different tree topologies were recovered depending on the molecular marker used. On one hand, the mitochondrial *cytochrome oxidase I* (*COI*) and the X-linked *period* gene

supported the hypothesis of two main clades, one including *D. buzzatii* and *D. koepferae* and another comprising the remaining species [48, 53, 54]. On the other hand, trees based on a few nuclear and mitochondrial markers supported the hypothesis that *D. koepferae* was sister to the *serido* sibling set [26, 55]. To further complicate this issue, not all the same species were analyzed in these studies. In this vein, a recent genomic level study using a large transcriptomic dataset supports the placement of *D. koepferae* in the *serido* sibling set and *D. buzzatii* as sister to this set [50] similar to the results of [26]. However, phylogenetic relationships within the *serido* sibling set could not be ascertained despite the magnitude of the dataset employed by Hurtado and co-workers [50]. Thus, our aim is to shed light on the evolutionary relationships within the *buzzatii* cluster by means of a mitogenomic approach.

In this paper, we report the assembly of the complete mitogenomes of *D. antonietae*, *D. borborema*, *D. buzzatii*, *D. seriema* and two strains of *D. koepferae*, together with the corresponding gene annotations. Unfortunately, *D. gouveai* and *D. serido*, that inhabit Brazilian arid lands, could not be included because they are difficult to obtain and are not available in *Drosophila* repositories. We also present a mitogenomic analysis that defines a different picture of the relationships within the *buzzatii* cluster with respect to the results generated with nuclear genomic data. Finally, we discuss possible causes of the discordance between nuclear and mitochondrial datasets.

## Material and methods

### Species selection

The mitochondrial genomes of six isofemale lines of five species of the *buzzatii* cluster were assembled for the present study, for which NGS data were available. *D. antonietae* (Dato) was collected in March 2010 on Martín García Island (Buenos Aires Province, Argentina 34˚10′42.12″S 58˚15′23.15"W) by J. Hurtado and E. Hasson. *D. borborema* (Dbrb) was obtained from the Drosophila Species Stock Center (Stock Number: 15081–1281.01, University of California, San Diego, USA) and derived from collections performed in 1974 on Morro do Chapéu (Bahía State, Brazil 11˚34′51.98″S 41˚07′08.13″W) by M. Wasserman and R.H. Richardson. *D. buzzatii* (Dbuz) was collected in summer 2010 in Lavalle (Mendoza Province, Argentina 32˚37′26.44″S 67˚34′15.20″W) by J. Hurtado and E. Hasson. Two *D. koepferae* (Dkoe) strains; strain *B* from collections made in Bolivia in December 1982 by A. Fontdevila and A. Ruiz, and strain *A* collected in Vipos (Tucumán, Argentina 26˚28′59″S 65˚22′00″W) in February 2010 by J. Hurtado and E. Hasson. *D. seriema* (Dsei, strain D73C3) derived from collections on Cachoeira do Ferro Doido (Bahía State, Brazil 11˚37' 40" S 41˚9' 2" W) in June 1990 by G. Kuhn and F.M. Sene [56]. The stocks of *D. antonietae*, *D. borborema*, *D. buzzatii* and *D. koepferae* are available upon request. The rationale of including these *D. koepferae* strains is motivated by previous protein electrophoresis work showing higher genetic divergence between Bolivian and Argentinian populations than between conspecific populations in other species [45]. In addition, we also included four species of the subgenus *Drosophila*, for which assembled mitogenomes were available as outgroups in the phylogenetic analyses: *D. grimshawi* (GenBank: BK006341.1), *D. littoralis* (GenBank: NC_011596.1), *D. virilis* (GenBank: BK006340.1) and *D. mojavensis* (GenBank: BK006339.1).

### In silico mtDNA reads extraction

Whole genome sequencing (WGS) and RNA-seq data for *D. antonietae*, *D. borborema* and both strains of *D. koepferae* were generated in our laboratory [34, 50]. For *D. seriema* and *D. buzzatii*, mitochondrial reads were retrieved from the Genome sequencing of *D. seriema* deposited in Sequence Read Archive database (SRA accession ID: ERX2037878) [56] and the

*D. buzzatii* genome project (https://dbuz.uab.cat), respectively. For each species, mitochondrial reads were extracted from genomic and transcriptomic (when available) datasets. Bowtie2 version 2.2.6 [57] was first used with default parameters (end-to-end sensitive mode) to map reads to the mitochondrial genome of *D. mojavensis*, the closest relative of *buzzatii* cluster species available, as a reference. Next, only reads that correctly mapped to the reference genome were retained using Samtools version 1.8 [58]. Finally, mapped reads from genomic and transcriptomic datasets were combined to generate a set of only mitochondrial reads.

## Mitochondrial reference genome assembly

It is well known that at least 25% of NGS reads are of mitochondrial origin [3]. Therefore, after the mapping process it is possible to attain a coverage ranging from 2000x to more than 20000x for mitogenomes. In order to avoid misassemblies caused by a large number of reads and given the difficulty of determining the coverage (and combination of reads) that recovers the complete mitochondrial genome, we split the reads into several datasets with different coverages by random sampling. Then, a three-step assembly procedure was adopted for these datasets based on recommendations of MITObim package version 1.8 [1]. In the first step, each dataset was employed to build a template by mapping its reads to the mitogenome of *D. mojavensis* using MIRA assembler [59]. In this way, several templates, based mostly on conserved regions, were built for each species. In the second step, entire mitogenomes were assembled by mapping the complete set of reads to the templates generated in the first step (coverage assembly), individually. This step was performed with the MITObim script and a maximum of ten mapping iterations. Finally, all the different coverage assemblies of the same species were aligned with Clustalw2 version 2.1 [60], and a consensus assembly was then considering a sequence representation threshold of 60% (nucleotide mostly represented at each position between assemblies), and not allowing gaps. De novo assemblies for each species, though more fragmented, were aligned to the assemblies obtained as described above and revealed the same gene order along the mitogenomes.

## PCR amplification, sanger sequencing and consensus correction

Mitogenome assembly coverage averaged more than 20000x; however, three regions including parts of *COI*, *NADH dehydrogenase subunit 6* (*ND6*) and *large ribosomal RNA* (*rRNAL*) genes showed low read representation in all species, producing miss-assemblies and fragmentation. These regions were PCR-amplified with GO taq Colorless Master Mix by Promega using primers designed for regions conserved across the six mitogenomes assembled in this study (data in S1 Text). PCR amplifications included an initial denaturation at 94˚C for 90 s, followed by 25 cycles of denaturation at 94˚C for 45 s, annealing at 62˚C for 50 s, extension at 72˚C for 1 min and a final 4 min extension. PCR fragments were sequenced in both directions on an ABI-3130xl (Genetic Analyzer). Sequences were analyzed and filtered using Mega X software [61] and, finally, merged with the assemblies.

## Genome annotation and bioinformatic analyses

The six new assemblies were annotated with the MITOS web server (http://mitos.bioinf.uni-leipzig.de) [62] using the invertebrate mitochondrial genetic code and default parameter settings. The position and orientation of annotations were examined by mapping reads to mitogenomes with Bowtie2 [57] and visualization conducted with IGV ver. 2.4.10 [63]. In addition, nucleotide composition and codon usage were analyzed using MEGA X [61]. A homemade python package (available upon request) was developed to compute the number of pairwise nucleotide differences in the *buzzatii* cluster, and to visualize its variation along the

mitogenomic alignment. Then we used the p-distance as a measured of nucleotide divergence, by dividing the number of nucleotide differences by the total number of nucleotides compared and by the number of pairwise comparisons [61]. Similar p-distance estimates were computed for the *D. melanogaster* subgroup with the aim to compare divergence patterns along the mitochondrial DNA in the *buzzatii* cluster with a deeply studied ensemble of species. To this end, one mitogenome of each one of the following species: *D. melanogaster* (KJ947872.2), *D. erecta* (BK006335.1), *D. simulans* (NC_005781), *D. sechellia* (NC_005780) and *D. yakuba* (NC_001322.1) were aligned, and nucleotide divergence estimates (p-distance) were obtained as described above. Synonymous ($d_S$) and non-synonymous substitution rates ($d_N$) were also estimated for each mitochondrial protein coding gene (PCG) using PAML 4.8 [64]. These estimates, as well as the ω ratio ($d_N/d_S$), were obtained separately for the *buzzatii* cluster and the *melanogaster* subgroup sequence alignments. Multiple sequence alignments of each coding gene were obtained with Clustalw2 ver. 2.1 [60].

## Phylogenetic analyses

Phylogenetic analyses were conducted considering PCGs, ribosomal genes (rRNAs), transfer RNA genes (tRNAs) and intergenic regions (excluding the control region) of the 6 mitogenomes plus the sequences of the outgroups *D. virilis*, *D. grimshawi*, *D. littoralis* and *D. mojavensis* (see details in species selection section). An alignment of the ten mitogenomes was performed with Clustalw2 version 2.1 [60]. The flanking sequences that correspond to the control region and portions of the alignment showing abundant gaps were manually removed with Seaview ver. 4 [65]. The final alignment was used as input in PartitionFinder2 [66] to determine the best partition scheme and substitution models, considering separate loci and codon position (in PCGs), which were used in Bayesian Inference and Maximum Likelihood phylogenetic searches. In the Bayesian Inference approach executed with MrBayes ver. 3.2.2 [67], both substitution model and parameter estimates were unlinked. Then, two independent Markov Chain Monte Carlo (MCMC) were run for 30 million generations with three samplings every 1000 generations, for a total of 30,000 trees. Tracer ver. 1.7.1 [68] was used to assess the convergence of the chain mixing, where all parameters had effective sample sizes (ESS) > 200, and 25% of the trees were discarded as burn-in and the remaining trees were used to estimate a consensus tree and the posterior probability of each clade. The consensus tree was plotted and visualized with FigTree ver. 1.4.4 (https://github.com/rambaut/figtree/releases) [69]. Maximum Likelihood searches were performed in 2,000 independent runs using RAxML ver. 8.2.11 [70], applying the rapid hill climb algorithm and the GTR+GAMMA model, considering the partition scheme obtained with PartitionFinder2. Two thousand bootstrap replicates were run to obtain clade frequencies that were plotted onto the tree with highest likelihood. Tree and bootstrap values were visualized with FigTree ver. 1.4.4 [69]. Bayesian Inference searches for each PCG were individually performed to identify correlations with the topology recovered using the complete mitogenome. The GTR-GAMMA model together with the same parameters and evaluation detailed before were applied on each MCMC.

## Divergence time estimation

Divergence times were estimated using the same methodology as in Hurtado et al. [50]. Four-fold degenerate third codon sites, i.e. putative neutral sites, of PCGs were extracted from the alignment and Bayesian Inference searches were analyzed with BEAST ver. 1.10.4 [71]. A strict clock was set using a prior for the mutation rate of $6.2 \times 10^{-07}$ per year (standard deviation of $1.89 \times 10^{-07}$), as was empirically estimated for mitochondrial DNA in *Drosophila melanogaster* [72]. In addition, a birth-death process with incomplete sampling and a time of 11.3 million

years ago -MYA- (CI = ~9.34 to ~13) [26] to the root were defined as tree priors. Two MCMC were produced in 30 million generations with tree sampling every 1000 generations. Tracer [68] was used to evaluate chain convergence, discarding 10% of the all trees (burn-in). The information of the recovered trees was summarized in one tree applying LogCombiner and TreeAnnotator ver. 1.10.4 (available as part of the BEAST package), including the posterior probabilities of the branches, the age of the nodes, and the posterior estimates and HPD limits of the node heights. The target tree was visualized using FigTree [69]. Only *D. mojavensis* was included as an outgroup in this analysis to minimize problems of among-taxa rate variation given by the large divergence between the *buzzatii* cluster and the rest of the species already sequenced, together with the lack of time point calibrations and accurate mutation rates.

## Results

### Mitogenome characterization, nucleotide composition and codon usage

The length of the assembled mitogenomes varied from 14885 to 14899 bp among the six strains reported in this paper. Mitogenomes consisted of a conserved set of 37 genes, including 13 PCGs, 22 tRNAs and 2 rRNAs genes, with order and orientation identical to *D. mojavensis*. Several short non-coding intergenic regions were also found. Twenty-three genes were found on the heavy strand (+) and fourteen on the light strand (-). Detailed statistics about metrics and composition of the mitogenomes are shown in Table 1.

Overall nucleotide composition in PCGs ranged between 37.6–37.8% A, 37.2–37.9% T, 10.2–10.4% G, and 14.1–14.7% C. The thirteen PCGs were AT-biased as in the entire mitogenome, and the codon usage bias in each gene was greater than 0.50. The most frequently used codons were UUA (Leu), AUU (Ile), and UUU (Phe) in all cases. Codon usage information for each species is shown in Table in S1 Table.

### Genetic diversity among mitogenomes

Patterns of divergence (p-distance) along the entire mitogenomes were, overall, very similar for both the *buzzatii* cluster and the *melanogaster* subgroup (Fig 2). However, overall divergence was higher in the *melanogaster* subgroup than in the *buzzatii* cluster. This difference between both ensembles is probably due to depth of divergence, the *melanogaster* subgroup comprises pairs of highly diverged species such as *D. melanogaster* and *D. yakuba* that split more than 10 MYA [73]. Despite the general similarity in the patterns of divergence, a substantial difference was found in the region encompassing *COIII*, *tRNA-G* and *ND3* genes. In this

**Table 1. Composition of mitochondrial elements in the species assemblies of the *Drosophila buzzatii* cluster.**

| Composition | Species assembly | | | | | | |
|---|---|---|---|---|---|---|---|
| | *D. antonietae* | *D. borborema* | *D. buzzatii* | *D. koepferaeB* | *D. koepferaeA* | *D. seriema* | *D. mojavensis* |
| Total length | 14885 | 14889 | 14889 | 14892 | 14891 | 14891 | 14904 |
| GC (%) | 23.36 | 23.22 | 23.60 | 23.20 | 23.20 | 23.26 | 23.54 |
| N's (%) | 0.00 | 0.00 | 0.00 | 0.00 | 0.00 | 0.00 | 0.00 |
| Intergenic (%) | 2.76 | 2.71 | 2.64 | 2.80 | 2.77 | 2.90 | 0.93 |
| tRNAs (%) | 9.82 | 9.81 | 9.80 | 9.81 | 9.81 | 9.80 | 9.78 |
| PCGs (%) | 72.98 | 73.04 | 73.15 | 72.95 | 72.97 | 72.85 | 75.15 |
| rRNAs (%) | 14.44 | 14.45 | 14.41 | 14.44 | 14.44 | 14.44 | 14.14 |

GC, Guanine+Cytosine; N's: undetermined nucleotides; Intergenic, regions between genes (non-coding); tRNAs, transfer RNA; PCGs, Protein Coding Genes; rRNAs, ribosomal RNA.

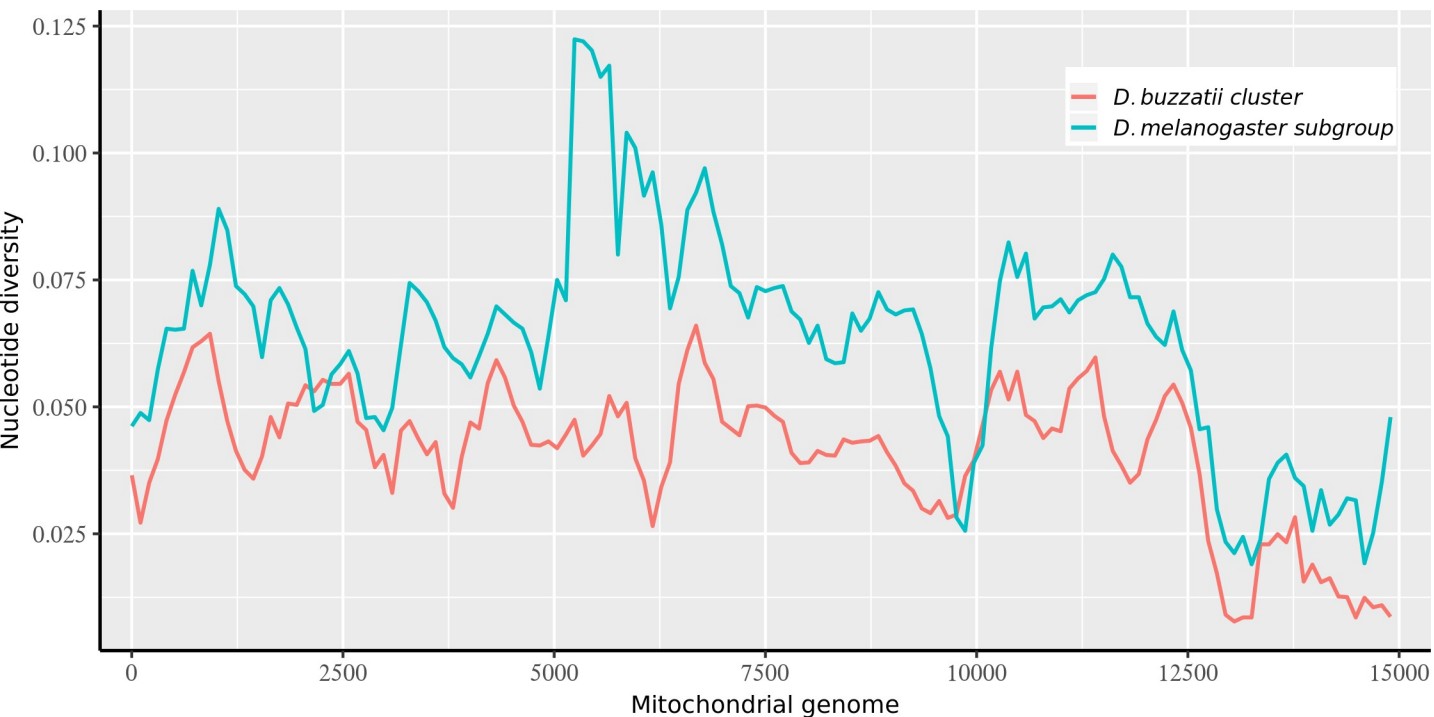

**Fig 2. Nucleotide diversity variation along the mitogenome, estimated for a sliding window of 500bp with an overlap of 100bp.** p-distance values for species belonging the *buzzatii* cluster and the *melanogaster* subgroup are represented independently.

region, from position 5000 to 6000, nucleotide diversity was the highest in the *melanogaster* subgroup, showing an apparent increase represented by two high peaks absent in the *buzzatii* cluster. Considering genetic divergence within the *buzzatii* cluster (Fig 3), the lowest value of average pairwise nucleotide divergence was observed for the pair *D. borborema* and *D. seriema* (p = 1.91x10$^{-03}$), while between *D. seriema* and *D. buzzatii* divergence was an order of magnitude larger (p = 2.73x10$^{-02}$). Divergence between *D. koepferae* strains was surprisingly high (7.14x10$^{-03}$). The complete set of divergence estimates in the *buzzatii* cluster is reported in Table in S2 Table. Substitution rates in synonymous ($d_S$) and non-synonymous ($d_N$) sites, and the ω ratios varied among PCGs (Table 2). The $d_N/d_S$ ratio (ω) varied from 0.003 to 0.060 among PCGs in the *buzzatii* cluster. The range in the *melanogaster* subgroup was similar, but with a lower upper bound (0.003–0.018). Two genes, *ATP8* and *ND2*, appear as outliers in the *buzzatii* cluster (*ATP8* and *ND2*), which apart from these two loci, had lower divergence values than in the *melanogaster* subgroup. The elevated $d_N/d_S$ values observed for *ATP8* and *ND2* in the buzzatii cluster (Table 2) are probably real since the true value of $d_S$ may be slightly underestimated due to multiple hits in the melanogaster subgroup (see above), leading to a slight overestimation of the $d_N/d_S$ ratio. In any case, these results suggest that purifying selection imposes strong constraints in the evolution of mitochondrial genes.

## Phylogenetic analyses

The sequences of the 13 PCGs, 22 tRNAs genes, 2 rRNAs genes, and intergenic regions were included in the alignment. Total length of the final matrix encompassing the ten mitogenomes was 15044 characters, from which 1950 were informative sites, 11583 conserved, and 1422 were singletons. Both Maximum Likelihood and Bayesian Inference phylogenetic analyses recovered the same highly supported topology that confirmed the

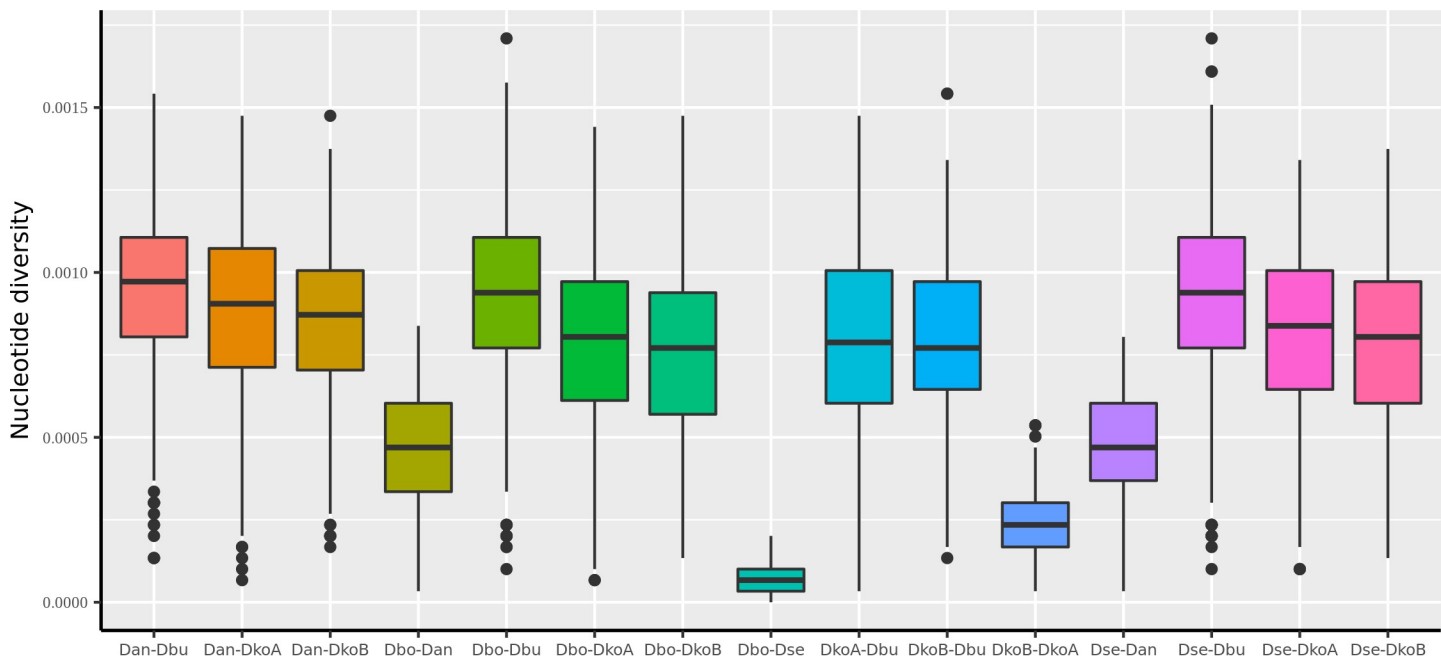

**Fig 3. Pairwise comparison of nucleotide diversity between species belonging the *buzzatii* cluster.** The official FlyBase abbreviations for *Drosophila* species names are shown.

monophyly of the *buzzatii* cluster (Fig 4). Two main clades can be observed in the tree, one including both *D. koepferae* strains as sisters to *D. buzzatii*, and the second, comprising *D. antonietae* as sister species of the sub-clade formed by *D. borborema* and *D. seriema*. The species selected as outgroups were placed as expected, with *D. mojavensis* as the closest relative of the *buzzatii* cluster. We also performed a gene tree analysis using all PCGs (S1 Fig). We could only obtain trees for seven genes out of the thirteen PCGs, given the

**Table 2. Estimates of non-synonymous ($d_N$) and synonymous ($d_S$) substitutions and their ratio (ω) among species of the *buzzatii* cluster and the *melanogaster* subgroup.**

| PCG | *buzzatii* cluster | | | *melanogaster* subgroup | | |
|---|---|---|---|---|---|---|
| | ω | $d_N$ | $d_S$ | ω | $d_N$ | $d_S$ |
| ATP6 | 0.005 | 0.007 | 1.390 | 0.014 | 0.034 | 2.364 |
| ATP8 | 0.060 | 0.031 | 0.506 | 0.014 | 0.071 | 4.892 |
| CytB | 0.005 | 0.008 | 1.463 | 0.010 | 0.027 | 2.556 |
| COI | 0.003 | 0.003 | 1.211 | 0.003 | 0.006 | 2.383 |
| COII | 0.005 | 0.005 | 1.058 | 0.008 | 0.018 | 2.295 |
| COIII | 0.006 | 0.008 | 1.277 | 0.012 | 0.024 | 1.975 |
| ND1 | 0.003 | 0.012 | 4.332 | 0.006 | 0.028 | 4.490 |
| ND2 | 0.036 | 0.040 | 1.102 | 0.016 | 0.039 | 2.368 |
| ND3 | 0.009 | 0.034 | 3.657 | 0.016 | 0.051 | 3.085 |
| ND4l | 0.008 | 0.009 | 1.158 | 0.001 | 0.013 | 8.026 |
| ND4 | 0.011 | 0.013 | 1.253 | 0.009 | 0.040 | 4.306 |
| ND5 | 0.007 | 0.018 | 2.564 | 0.013 | 0.063 | 4.764 |
| ND6 | 0.012 | 0.027 | 2.231 | 0.018 | 0.082 | 4.534 |

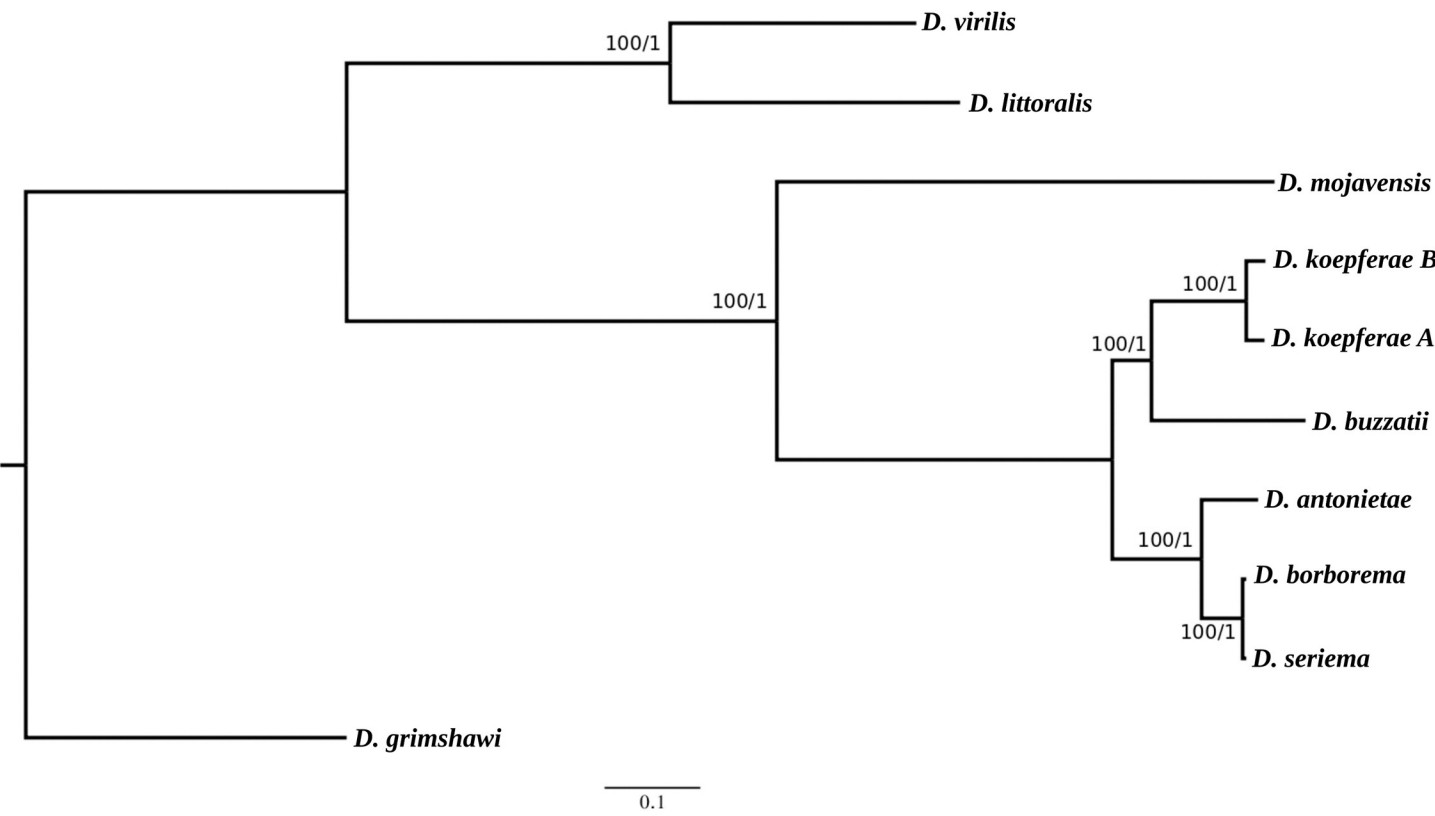

**Fig 4. Phylogenetics hypotheses for the *buzzatii* cluster based on the entire sequence of the mitogenome (control region not included).** Tree topology recovered by both Maximum Likelihood and Bayesian Inference searches. Bootstrap values and posterior probabilities are indicated at each node.

lack of informative sites in the alignments of *ATP8*, *ATP6*, *ND3*, *ND4l*, *COII* and *COIII*. Only two (*ND1* and *ND5*) out of the seven recovered gene trees showed the same topology as the complete mitogenome, while the remaining genes produced three (different) topologies. Trees obtained with *CytB* and *ND4* showed *D. buzzatii* as sister to the *serido* sibling set which included *D. koepferae*. *COI* and *ND2* retrieved trees where *D. buzzatii* and *D. koepferae* exchanged positions in the tree, placing *D. koepferae* as the species closest to the putative ancestor of the cluster. *ND6* recovered two clades where *D. antonietae* was the sister of *D. buzzatii* and *D. koepferae* (both strains) in one clade, and the pair *D. borborema*-*D. seriema* composed the other.

## Divergence times

PCGs contained 1201 4-fold degenerate sites in the mitogenomes of the *buzzatii* cluster strains assembled in this study. The tree obtained in the divergence time estimation analysis (Fig 5) was topologically identical to the trees obtained in the phylogenetic analyses using complete mitogenomes (see Fig 4). Divergence time estimations showed that the *buzzatii* cluster diverged in the Early Pleistocene, 2.11 MYA, and the split with the *D. mojavensis* common ancestor occurred 10.63 MYA in the Miocene. Our results also indicated that the clade containing *D. antonietae*, *D. borborema* and *D. seriema*, is younger than the clade composed by *D. buzzatii* and *D. koepferae*. In addition, the split between *D. borborema* and *D. seriema* is quite recent, about ~50,000 years ago, in the Late Pleistocene, even more recent than the split of *D. koepferae* strains that diverged ~310,000 years ago, in the Middle Pleistocene.

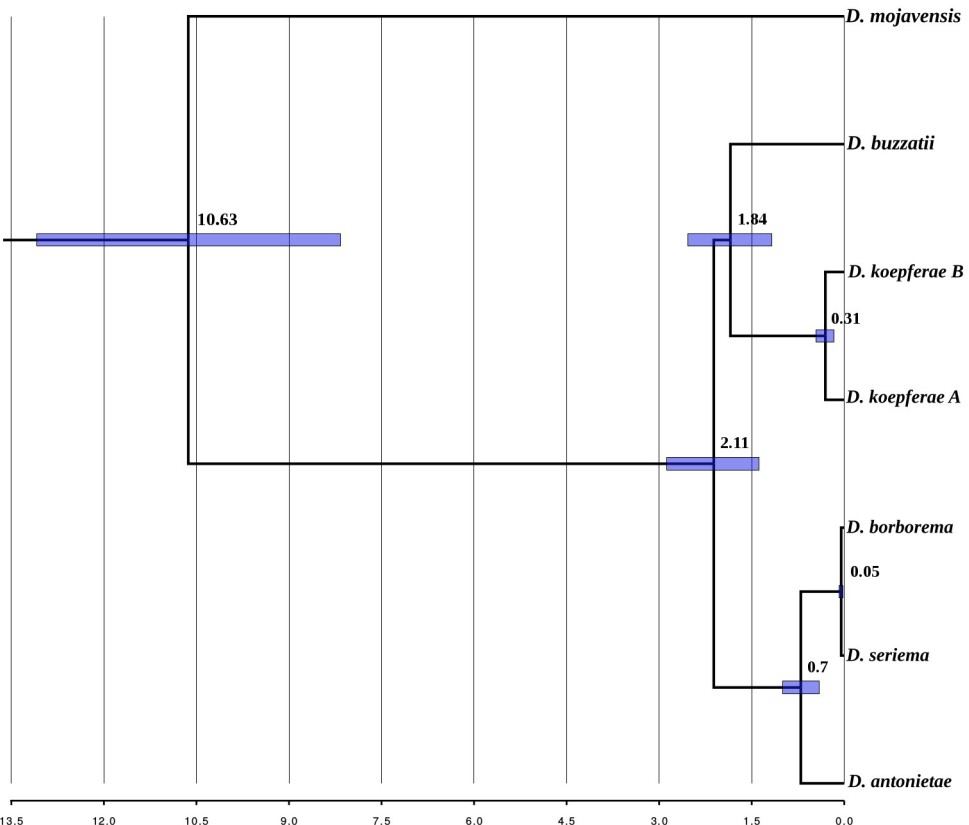

**Fig 5. Divergence times for the *buzzatii* cluster drawn on a Bayesian inference tree.** Numbers on each node are the time estimates. Blue bars represent the 95% confidence intervals of estimates.

## Discussion

The six newly assembled mitochondrial genomes of five cactophilic species of the *buzzatii* cluster share molecular features with animal mitochondrial genomes sequenced so far [74]. All assembled mitogenomes contain the same set of genes usually found in animal mitochondrial genomes. Gene order and orientation, as well as the distribution of genes on the heavy and light strands were identical to the mitogenome of the closest relative *D. mojavensis* and other drosophilids [9]. Analysis of overall nucleotide composition of mitogenomes and PCGs revealed the typical AT-bias found in *Drosophila* mitogenomes. Codon usage is highly biased suggesting that synonymous sites cannot be considered strictly neutral and that some sort of natural selection for translational accuracy governs codon usage [75].

Phylogenetic analyses based either on complete mitogenomes or four-fold degenerate sites (for divergence time estimations), retrieved a well supported tree, suggesting that the cluster is composed by two main clades, one including *D. buzzatii* and *D. koepferae* (both strains) and another comprising *D. antonietae*, *D. seriema* and *D. borborema*. Though our present results are consistent with previous work based on single mitochondrial genes [53, 76], they should be considered with caution since we only included a single inbred line as representative of each species, except for *D.* koepferae. Moreover, phylogenetic relationships depicted by mitogenomes do not agree with phylogenetic studies based on both a small set of nuclear and mitochondrial genes [26] and a large set of nuclear genes -see below- [50]. Interestingly, the topology showing these two clades was only recovered in two (*ND1* and *ND5*) out of seven

trees based on individual PCGs, and the remaining gene trees produced either a novel topology or a topology consistent with the phylogeny reported in Hurtado et al. [50].

The lack of recombination causes mitochondrial DNA to be inherited as a unit, so trees recovered with individual mitochondrial genes are expected to share the same topology and to be consistent with the trees obtained with complete mitogenomes. On the other hand, our results suggest that individual genes not only produce different topologies but also a poor resolution of phylogenetic relationships. Such inconsistencies between complete mitogenomes and gene trees in phylogenetic estimation may result from inaccurate reconstruction or from real differences among gene trees. The first possible explanation is the simple fact that numbers of informative sites within a single locus are insufficient to accurately estimate phylogenetic relationships, particularly in groups of recently diverged species: overall support for gene trees was poorer than for the tree based on complete mitogenomes. Second, heterogeneity in evolutionary rates among genes and/or differences in selective constraints along the mitogenome can also account for these inconsistencies [17, 77–79] consistent with our observations of substantial variation in synonymous and nonsynonymous rates as well as ω ratios across PCGs. In addition, variation among oxidative phosphorylation complexes in the *buzzatii* cluster was high. The *ND* complex was, on average, less constrained than the *ATP* complex, *cytochrome b* (*CytB*) and *cytochrome oxidase* complex (*COI*, *COII & COIII*), consistent with results reported in the *melanogaster* subgroup [9, 80]. Another factor that may lead to biased tree construction, particularly relevant for mitochondrial genes characterized by high substitution rates, is substitutional saturation [81]. A priori, saturation should not be problematic in recently diverged species, like the *buzzatii* cluster, however, saturation may be problematic in the estimation of divergence relative to the outgroup and, thus, for phylogenetic inference. The closest outgroup to the *buzzatii* cluster employed in our study was the *mulleri* complex species *D. mojavensis*. Available evidence suggest that these complexes diverged ~10 MYA [26], but possibly more recently, ca 5.5 MYA [50] suggesting that substitution saturation may lead to inaccurate phylogenetic reconstruction.

In this context, a recent report investigating the effect of using individual genes, subsets of genes, complete mitogenomes and different partitioning schemes on tree topology suggested a framework to interpret the results of mitogenomic phylogenetic studies [11]. The authors concluded that trees obtained with complete mitogenomes reach the highest phylogenetic performance and reliability than single genes or subsets of genes. Therefore, we consider that phylogenetic relationships inferred from complete mitogenomes reflect the evolutionary history of, at least, mitogenomes.

The phylogenetic relationships depicted by our mitogenomic approach are incongruent with a recent study based on transcriptomic data [50]. Based on a concatenated matrix of 813 kb uncovering 761 gene regions, the authors obtained a well-supported topology in which *D. koepferae* appears phylogenetically closer to *D. antonietae* and *D. borborema* than to *D. buzzatii*, placing *D. buzzatii* alone as sister to the rest of the cluster. This topology agrees with male genital morphology, cytological and molecular phylogenetic evidence [26, 35, 55]. Nevertheless, the pattern of cladogenesis of the trio *D. koepferae-D. borborema-D. antonietae* could not be fully elucidated since a nuclear gene tree analysis yielded ambiguous results. As a matter of fact, the analysis of the 761 gene trees reported showed that about one third of the genes supported each one of the three possible topologies for the trio *D. koepferae-D. antonietae-D. borborema* indicating a hard polytomy [50]. In contrast, the early separation of *D. buzzatii* from the *serido* sibling set is supported by 97% of the gene trees and, surprisingly, none of the gene trees recovered the clade including *D. buzzatii* and *D. koepferae* as the sister group of the clade involving *D. antonietae* and *D. borborema* [50] as suggested by the present mitogenomic approach. Such mitonuclear discordance has been reported in several animal species. A recent review lists several examples in animals [82]. Likewise, the literature in this respect is abundant

in the genus *Drosophila*. Well-known cases are *D. pseudoobscura* and *D. persimilis* [83]; *D. santomea* and *D. yakuba* [84]; and *D. simulans* and *D. mauritiana* [85]. Mitonuclear discordance may be caused by incomplete lineage sorting (ILS) and/or introgressive hybridization. These two factors do not equally affect mitochondrial and nuclear genomes because ILS is more likely for nuclear genes, especially when the ancestral effective population size of recently diverged species was large [86, 87]. Introgressive hybridization is expected to be prevalent in mitochondrial genomes given its lower effective population size [88]. If we accept that the topology based on nuclear genes is representative of the species-history (see also [49]), the greater similarity between *D. buzzatii* and *D. koepferae* mitogenomes is suggestive of gene flow between these largely sympatric species [35]. Thus, we suggest that *D. buzzatii* and *D. koepferae* lineages initially separated but then exchanged genes via fertile F1 females (males were likely sterile as expected due to Haldane's Rule) before finally separating less than 1.5 MYA. Not only the more recent mitogenomic ancestry is suggestive of gene exchange, also traces of introgressive hybridization can still be detected in nuclear genomes [50].

In fact, phylogenetic, population genetic and experimental hybridization studies suggest a significant role of introgression in the evolutionary history of the *buzzatii* cluster. Phylogeographic studies revealed discordances between mitochondrial markers and genital morphology in areas of sympatry between species [53]. Likewise, interspecific gene flow has been invoked to account for shared nucleotide polymorphisms in nuclear genes in *D. buzzatii* and *D. koepferae* that cannot be accounted by ILS [89, 90]. Moreover, experimental hybridization studies have shown that several species of the *buzzatii* cluster can be successfully crossed, producing fertile hybrid females that can be backcrossed to both parental species. Interestingly, *D. koepferae* can be successfully crossed with *D. antonietae*, *D. borborema*, *D. buzzatii* and *D. serido* [45, 48, 91–95].

Our estimates of divergence times are in conflict with most previous studies. In general, previous estimates, based on individual genes or a few genes (either mitochondrial or nuclear) suggested an older origin of the cluster and deeper splitting times within the cluster when compared to the estimates based on transcriptomes and mitogenomes. In effect, Gómez & Hasson [89] and Oliveira et al. [26] dated the split of *D. buzzatii* from the remaining species of the cluster at 4.6 MYA, respectively, whereas Manfrin et al.'s estimates [48] were even older, from 3 to 12 MYA for the most recent to the more ancient split. In contrast, putting apart the divergence time of the clade *D. buzzatii-D. koepferae* for the reasons discussed above, the radiation of the remaining three species seems to be extremely recent, less than 1 MYA using mitochondrial genomes (Fig 5), which are similar to estimates based on transcriptomes [50]. However, it is worth mentioning that divergence times estimated in the present paper and by Hurtado et al. [50] may be biased downwards since both are based on empirical mutation rates for nuclear and mitochondrial genes, respectively, calculated for over 200 generations in *D. melanogaster* [72]. Thus, these results should be interpreted with caution in the light of evidence suggesting not only the time-dependence of molecular evolutionary rates, but also that mutation rates obtained using pedigrees and laboratory mutation-accumulation lines often exceed long-term substitution rates by an order of magnitude or more [79].

Even though divergence times estimates obtained in this study cannot be entirely compared to assessments based on nuclear genomic data and individual nuclear genes, given the uncertainty of tree topology, they concur in that species of the *buzzatii* cluster apparently emerged during the Late Pleistocene in association with Quaternary climate fluctuations [49, 50, 76]. Moreover, in view of the obligate ecological association between *buzzatii* cluster species and cacti, the so-called Pleistocene "refuge hypothesis" is a suitable explanation for the diversification in this group in active cladogenesis. This hypothesis suggests that Pleistocene glacial cycles successively generated isolated patches of similar habitats across which populations may have diverged into species [96, 97].

Available paleo-climatic evidence, consistent with the Pleistocene "refuge hypothesis", can also account for the relatively deep intraspecific divergence between Bolivian and Argentinian *D. koepferae* strains. Because Quaternary topographical patterns in the Central Andes have remained unchanged for the last 2–3 MYA, a plausible explanation for this late Pleistocene vicariant event is related with glacial-interglacial cycles [98]. Although the validity of the Pleistocene "refuge hypothesis" is controversial (cf. [99]) and few studies addressed specific hypotheses on how the Quaternary glacial-interglacial cycles impacted species diversification [100], our divergence time estimates between Bolivian and Argentinian *D. koepferae* suggest a role of climatic oscillations as a factor of ecogeographical isolation in the Central Andes during the Pleistocene. Moreover, paleo-climatological evidence suggest that the area inhabited by *D. koepferae* has been exposed to substantial climatic variations on timescales of $10^3$–$10^5$ years related with glacial-interglacial cycles. Thus, Andean north-south exchanges may have been alternately favored or disfavored by these Quaternary climatic oscillations. In fact, the estimated age of the vicariant event between the *D. koepferae* strains is tantalizingly coincident with the coldest phase of the Marine Isotopic Stage (MIS) 10, which corresponds to a glacial period that ended about 337,000 years ago [101]. The coldest period of the MIS 10, recorded in global air and sea surface temperature and also the lowest atmospheric $CO_2$ levels, occurred ca 355,000 years, well within the confidence interval of our divergence time estimated between *D. koepferae* strains. On a global scale, glacial periods are primarily reflected in a lowering of air temperature but also in altered patterns of precipitation in the both sides of the Central Andes [102] which were in turn the main drivers of vegetation changes [103] including the appearance of South American columnar cacti [104]. Besides the impact on air temperature, periods of ice advance in the Central Andes generally were periods of negative water balance in the Pacific coastal regions west to the Central Andes [105], and a positive water balance in the Central Andes, as evidenced by deeper and fresher conditions in Lake Titicaca [106] (see S2 Fig). Thus, during the colder and wetter phases of the MIS 10 in the Central Andes, species distributions may have suffered a general contraction towards the southern and northern lowland warmer refugia between 1000–2000 m, whereas a general worsening condition occurred in higher western elevations. Northern and southern refugia were probably separated by a gap of low suitability represented by the steep gradient of the eastern flank of Eastern Andes between 22–24˚S, which represents today a region of strong W-E precipitation gradient. The MIS 10 glacial cycle has a particular structure since it does not have a pronounced interstadial (relative warmer) conditions in the mid-cycle [107], providing a prolonged, effective "soft" dispersal barrier that affected the distribution of *D. koepferae*.

Finally, future analyses including the mitogenomes of the other Brazilian species *D. gouveai and D. serido* and several mitogenomes of each species are needed to achieve a more complete understanding of the evolutionary history of the cluster. Such comparative analysis including the complete mitogenomes of all *buzzatii* cluster species will help to disentangle the intricate relationships in this group.

## Supporting information

**S1 Text. Set of primers used to sequence each gene.**
(DOCX)

**S1 Table. Codon usage for each mitogenome of the *buzzatii* cluster species.**
(XLSX)

**S2 Table. Genetic divergence among species of the *buzzatii* cluster.** Estimates are shown for each pairwise comparison between species.
(XLSX)

**S1 Fig. Phylogenetic hypotheses for the *buzzatii* cluster species recovered by each mitochondrial gene using Bayesian Inference searches.**
(TIF)

**S2 Fig. Paleoclimatic records of the last 500,000 years.** Ages in the top are indicated as 103 years (kyrs). Gradated shading area indicates divergence age estimates. Marine Isotope Stages (MIS) are labeled according to Lisiecki and Raymo [101]. Shaded vertical areas correspond to glacial periods whereas white areas correspond to interglacials or interstadials. Glacial periods correspond to cold and dry conditions in the western slopes of the Western Andes, and cold and wetter conditions in the eastern slopes of the Eastern Andes and the Altiplano. **A**. Globally-averaged surface air temperature anomaly reconstructed from proxy and model data for the last eight glacial cycles [108]. **B.** $CO_2$ concentration based on Vostok Ice Core data [109]. **C.** Iron accumulation rates (AR Fe) reflecting changes in terrigenous sediment input to ODP Site 1239D, Equatorial Pacific [110]. **D.** % of $CaCO_3$ from Site LT01-2B indicating changes in water balance at Lake Titicaca Basin, Bolivia (modified from [106]).
(TIF)

## Acknowledgments

The authors wish to thank D. Rand, an anonymous reviewer and the academic editor W.J. Etges for useful comments and constructive criticisms that helped to improve previous versions of the manuscripts. NNM is a CONICET PhD student. JM, JH, FA and EH are members of Carrera del Investigador Científico y Tecnológico (CIC) of CONICET.

## Author Contributions

**Conceptualization:** Nicolás Nahuel Moreyra, Julián Mensch, Juan Hurtado, Francisca Almeida, Cecilia Laprida, Esteban Hasson.

**Data curation:** Nicolás Nahuel Moreyra, Julián Mensch.

**Formal analysis:** Nicolás Nahuel Moreyra, Julián Mensch, Juan Hurtado, Francisca Almeida.

**Funding acquisition:** Esteban Hasson.

**Investigation:** Nicolás Nahuel Moreyra, Juan Hurtado, Francisca Almeida, Cecilia Laprida, Esteban Hasson.

**Methodology:** Nicolás Nahuel Moreyra, Julián Mensch, Juan Hurtado, Francisca Almeida.

**Project administration:** Esteban Hasson.

**Resources:** Esteban Hasson.

**Software:** Nicolás Nahuel Moreyra.

**Supervision:** Julián Mensch, Francisca Almeida, Cecilia Laprida, Esteban Hasson.

**Validation:** Nicolás Nahuel Moreyra, Julián Mensch.

**Visualization:** Nicolás Nahuel Moreyra, Cecilia Laprida.

**Writing – original draft:** Nicolás Nahuel Moreyra, Esteban Hasson.

**Writing – review & editing:** Nicolás Nahuel Moreyra, Julián Mensch, Juan Hurtado, Francisca Almeida, Cecilia Laprida, Esteban Hasson.

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
