## [Decision Letter · Decision Letter 0]

30 Aug 2019

PONE-D-19-20340

What does mitogenomics tell us about the evolutionary history of the Drosophila buzzatii cluster (repleta group)

PLOS ONE

Dear Mr. Moreyra,

Thank you for submitting your manuscript to PLOS ONE. After careful consideration, we feel that it has merit but does not fully meet PLOS ONE’s publication criteria as it currently stands. Therefore, we invite you to submit a revised version of the manuscript that addresses the points raised during the review process.

Two external referees and me have now evaluated your paper. Both reviewers raised some very important issues that need to be addressed in a revision. I re-read the MS and found a large number of grammar and translation issues that must be corrected satisfactorialy in a revised MS. I have uploaded a marked up manuscript to aid you in the revision process. I have made many comments that need to be addressed in this marked up manuscript, but I may have missed some that need fixing. Please proofread carefully.

A few other points need explanation, such as why some in-group species were not included in this analysis, and some cautionary explanation about the use of single inbred lines to represent these species, except for *D. koepferae*.  I'd guess that more geographically disparate samples would complicate this analysis and change your conclusions as well. Further, using a "total evidence" approach, why not use all available genetic data including inversions to build these trees?  I don't think its surprising that mtDNA yields different results than nuclear markers. However, the acquisition of more mitogenomes for these species is a valuable exercise.

We would appreciate receiving your revised manuscript by Oct 14 2019 11:59PM. To enhance the reproducibility of your results, we recommend that if applicable you deposit your laboratory protocols in protocols.io, where a protocol can be assigned its own identifier (DOI) such that it can be cited independently in the future. For instructions see: http://journals.plos.org/plosone/s/submission-guidelines#loc-laboratory-protocols

We look forward to receiving your revised manuscript.

Kind regards,

William J. Etges

Academic Editor

PLOS ONE

Journal Requirements:

2. In your Methods section, please provide additional location information of the collection sites, including geographic coordinates for the data set if available.

4. We note that Figure 1 in your submission contain map/satellite images which may be copyrighted. All PLOS content is published under the Creative Commons Attribution License (CC BY 4.0), which means that the manuscript, images, and Supporting Information files will be freely available online, and any third party is permitted to access, download, copy, distribute, and use these materials in any way, even commercially, with proper attribution. For these reasons, we cannot publish previously copyrighted maps or satellite images created using proprietary data, such as Google software (Google Maps, Street View, and Earth). For more information, see our copyright guidelines: http://journals.plos.org/plosone/s/licenses-and-copyright.

You may seek permission from the original copyright holder of Figure 1 to publish the content specifically under the CC BY 4.0 license.

If you are unable to obtain permission from the original copyright holder to publish these figures under the CC BY 4.0 license or if the copyright holder’s requirements are incompatible with the CC BY 4.0 license, please either i) remove the figure or ii) supply a replacement figure that complies with the CC BY 4.0 license. Please check copyright information on all replacement figures and update the figure caption with source information. If applicable, please specify in the figure caption text when a figure is similar but not identical to the original image and is therefore for illustrative purposes only.The following resources for replacing copyrighted map figures may be helpful:

Reviewers' comments:

Reviewer's Responses to Questions

**Comments to the Author**

1. Is the manuscript technically sound, and do the data support the conclusions?

Reviewer #1: Yes

Reviewer #2: Yes

2. Has the statistical analysis been performed appropriately and rigorously? 

Reviewer #1: Yes

Reviewer #2: Yes

3. Have the authors made all data underlying the findings in their manuscript fully available?

Reviewer #1: Yes

Reviewer #2: Yes

4. Is the manuscript presented in an intelligible fashion and written in standard English?

Reviewer #1: Yes

Reviewer #2: Yes

5. Review Comments to the Author

Reviewer #1: Comments to MS by Moreyra et al. PLOS ONE

General comments

This paper reports the sequencing, assembly and annotation of six mitochondrial genomes of five Drosophila repleta group species (subgenus Drosophila). This is useful information because none of these mitogenomes were previously available. The authors analyze these genomes to characterize patterns of divergence along the mitogenome, to elucidate the phylogenetic relationships between the five species and to estimate divergence times among them. It seems to me a straightforward, well written and useful paper. My comments are relatively minor.

Specific suggestions

Please, when necessary (e.g. Figure 3) use the official FlyBase abbreviations for Drosophila species names.

Dato = D. antonietae

Dbrb = D. borborema

Dbuz = D. buzzatii

Dkoe = D. koepferae

Dsei = D. seriema

To refer to the two strains of D. koepferae you should use always the same name: either A and B, or 11 and 7.1. Currently there is a mixture, compare Figure 3 and Figure 4.

Page 2, lines 29-30. I suggest to use the alphabetical order for species names (as in Table 1): D. antonietae, D. borborema, D, buzzatii, D. koepferae (two stocks) and D. seriema.

Figure 3. I don’t see the logical in the order of the pairwise comparisons. Perhaps you could use the alphabetical order (in agreement with previous comment):

Dato – Dbrb

Dato – Dbuz

Dato – Dkoe A

Dato – Dkoe B

Dato – Dsei

Dbrb – Dbuz

Dbrb – Dkoe A

Dbrb – Dkoe B

Dbrb – Dsei

Dbuz – Dkoe A

Dbuz – Dkoe B

Dbuz – Dsei

Dkoe A – Dkoe B

Dkoe A – Dsei

Dkoe B – Dsei

Pages 6-7, lines 144-155. It is not clear to me whether the authors produced a “de novo” assembly of the six mitogenomes or they used D. mojavensis as a template. This is important in order to conclude (page 10) that PCGs in the six mitogenomes show the same order and orientation than in D. mojavensis. Please clarify this issue.

Page 11, lines 246-247. Please give more details on how pairwise nucleotide diversity (�) was calculated. Because you have six mitogenomes in the buzzatii cluster, there is a total of 15 pairwise comparisons. Is the value reported in Figure 2 the average of the 15 pairwise comparisons? If this is the case, this value includes both interspecific and intraspecific (two stocks of D. koepferae) comparisons. Clarify.

Minor corrections

Page 2, line 33. The monophyly of that the D. buzzatii cluster. Delete “that”

Page 5, line 108. Insert “cluster” after buzzatii

Page 6, line 134. Split the sentence:

… unpublished data). For D. seriema and D. buzzatii, mitochondrial reads were retrieved …

Page 13, line 283. We could only obtain trees for ….

Page 13, line 287. Allocated -> placed

Page 13, line 294. Legend of Fig 4. Bootstrap values and posterior probabilities are indicated at each node.

Page 15, line 334. Thus -> on the other hand,

Heading of Table S1. Set of primers used to sequence each gene.

Reviewer #2: Moreyra et al. report on the assembly and phylogenetic analysis of mtDNAs from members of the Drosophila buzzatii group. Only one of the members of this clade had prior mtDNA assembly data, so this is a contribution to the genetics of this group of species. Using available genomic sequence data, the mtDNAs were assembled using bioinformatics tools, and gaps in coverage were validated using direct Sanger sequencing of amplified regions. The paper presents some straightforward molecular evolution and phylogenetic analyses that are consistent with other Drosophila mitogenomic analyses, but reveal some differences with the melanogaster group. The data are concordant with various biogeographic scenarios, and reveal some discordance with nuclear phylogenetic analyses. Below are a few comments that may improve the manuscript.

Line 33. Delete extra ‘that’

Line 246: A terminology point: ‘pi’ for pairwise diversity usually refers to polymorphism within populations, and estimates this value from a random sample of alleles. The use of pi from a sample of species has a somewhat different interpretation given that taxon sampling is different from a random sample of alleles from a breeding population. The use of a pairwise metric is fine, but some clarification should be made in these different contexts.

Line 248: The pi values are larger in the melanogaster subgroup than in the buzzatii cluster. Related to the comment above, if these were equivalent random samples from two different idealized populations the difference would imply different population genetic forces, but since the differences in divergence times and distribution of taxa in the two subgroups / clusters being compared are not the same, the pi metric may simply reflect those historical contingencies. The metric pi is sensitive to the frequencies of alleles in a population, so more closely related mtDNAs will reduce pi, or more deeper-diverging mtDNAs will inflate it. So, some comment should be made to this effect to provide context for the larger pi values in the melanogaster subgroup. Notably, that group has three very diverged mtDNAs within the D. simulans clade that may inflate pi, which is not seen in the buzzatii sample. Clarify that the Dmel vs. Dbuzz difference is probably due to depth of divergence, not something distinct about rates of evolution. The use of a pairwise metric is fine, but its interpretation is complicated by the difference between a taxon sample vs. an allele sample from a breeding population.

Lines 261-263: The higher dN/dS values for two genes in Dbuzz vs. Dmel data (Table2), may imply a shift in function. It could be noted that since the dS values (and pi as per points above) is higher in Dmel group, the true value of dS may be slightly underestimated due to multiple hits, leading to a slight overestimation of the dN/dS ratio in Dmel. This means that the elevated values of dN/dS for the ND and ATP genes in the Dbuzz group relative to the same Dmel genes are probably real.

6. PLOS authors have the option to publish the peer review history of their article (what does this mean?). If published, this will include your full peer review and any attached files.

Reviewer #1: No

Reviewer #2: Yes: David Rand

---

## [Author Response · Author response to Decision Letter 0]

18 Sep 2019

Editorial Board

Dr. William J. Edges

Academic Editorial

Dear Dr. Edges,

 First, we would like to thank the subject editor Dr. W.J. Etges and the two reviewers, an anonymous one and Dr. D. Rand, for helpful and insightful comments that have definitely contributed to improve manuscript entitled “What does mitogenomics tell us about the evolutionary history of the Drosophila buzzatii cluster (repleta group)?” by Nicolás Moreyra et al.

 We revised the manuscript addressing all the issues raised by the academic editor and both reviewers. Below, we transcribe each one of the comments, followed by our response (Author´s response), which outlines the changes made to the manuscript in order to reflect the suggestions made in each case.

Responses to the Academic Editor

Two external referees and me have now evaluated your paper. Both reviewers raised some very important issues that need to be addressed in a revision. I re-read the MS and found a large number of grammar and translation issues that must be corrected satisfactorialy in a revised MS. I have uploaded a marked up manuscript to aid you in the revision process. I have made many comments that need to be addressed in this marked up manuscript, but I may have missed some that need fixing. Please proofread carefully.

Author’s response: all comments and suggestions in the manuscript marked up by the editor were taken into account and incorporated in the revised version.

A few other points need explanation, such as why some in-group species were not included in this analysis, and some cautionary explanation about the use of single inbred lines to represent these species, except for D. koepferae. I'd guess that more geographically disparate samples would complicate this analysis and change your conclusions as well. 

Author’s response: D. gouveai and D. serido are species that inhabit Brazilian arid lands, and could not be included in this study because of the unavailability of stocks in Drosophila repositories for these species. This is detailed in the revised manuscript. We included a sentence in the discussion section providing a cautionary note related to the use of single lines as representative of species, except for Dkoe, which was represented by two highly differentiated lines.

Further, using a "total evidence" approach, why not use all available genetic data including inversions to build these trees? I don't think its surprising that mtDNA yields different results than nuclear markers. However, the acquisition of more mitogenomes for these species is a valuable exercise.

Author’s response: We did not included other genetic data to build the trees because in a recent phylogenetic study based on a large nuclear genes dataset, considered other sources of information, like morphology and inversions. The aim of the present study was to compare the results of our mitogenomic approach with the study based on transcriptomic data.

Responses to reviewers

Reviewer #1:

Specific suggestions.

Please, when necessary (e.g. Figure 3) use the official FlyBase abbreviations for Drosophila species names.

Dato = D. antonietae

Dbrb = D. borborema

Dbuz = D. buzzatii

Dkoe = D. koepferae

Dsei = D. seriema

Author’s response: we adopted the official FlyBase abbreviations for the species names.

To refer to the two strains of D. koepferae you should use always the same name: either A and B, or 11 and 7.1. Currently there is a mixture, compare Figure 3 and Figure 4.

Author’s response: we corrected references to strains of D. koepferae to A and B (instead of 11 and 7.1) in the text, Figure 3, Figure 4 and Figure 5 (as well as in S1 Fig).

Page 2, lines 29-30. I suggest to use the alphabetical order for species names (as in Table 1): D. antonietae, D. borborema, D, buzzatii, D. koepferae (two stocks) and D. seriema.

Figure 3. I don’t see the logical in the order of the pairwise comparisons. Perhaps you could use the alphabetical order (in agreement with previous comment):

Dato – Dbrb

Dato – Dbuz

Dato – Dkoe A

Dato – Dkoe B

Dato – Dsei

Dbrb – Dbuz

Dbrb – Dkoe A

Dbrb – Dkoe B

Dbrb – Dsei

Dbuz – Dkoe A

Dbuz – Dkoe B

Dbuz – Dsei

Dkoe A – Dkoe B

Dkoe A – Dsei

Dkoe B – Dsei

Author’s response: We change the order of the species names in Table 1 to alphabetical order. However, we included D. mojavensis in the last column (avoiding the sorted list) as a comparative external reference.

Pages 6-7, lines 144-155. It is not clear to me whether the authors produced a “de novo” assembly of the six mitogenomes or they used D. mojavensis as a template. This is important in order to conclude (page 10) that PCGs in the six mitogenomes show the same order and orientation than in D. mojavensis. Please clarify this issue.

Author’s response: We produced a reference assembly only for the conserved regions between D. mojavensis and each species belonging the buzzatii cluster, creating several templates (based on different subsets of reads) for each species. After that, the complete set of mitochondrial reads (of each species) was mapped to the corresponding templates to generate several genome assemblies that were aligned to create a consensus per species. We also performed de novo assemblies for each species (line 166) that were more fragmented, but happily they revealed the same gene order along the mitogenomes. These details are clarified in the revised manuscript.

Page 11, lines 246-247. Please give more details on how pairwise nucleotide diversity (pi) was calculated. Because you have six mitogenomes in the buzzatii cluster, there is a total of 15 pairwise comparisons. Is the value reported in Figure 2 the average of the 15 pairwise comparisons? If this is the case, this value includes both interspecific and intraspecific (two stocks of D. koepferae) comparisons. Clarify.

Author’s response: we estimated nucleotide divergence by calculating the p-distance (analogous of pi), which is obtained by dividing the number of nucleotide differences between species pairs by the total number of nucleotides compared and by the number of pairs of sequences being compared (six in our study). We decided to independently include stocks of D. koepferae given the significant divergence between them, which is even greater than between D. borborema and D. seriema. In order to avoid misunderstandings regarding the parameter pi we decided to refers the measure of divergence as p-distance (instead of pi since this parameter usually refers to polymorphisms within populations).

Minor corrections

Page 2, line 33. The monophyly of that the D. buzzatii cluster. Delete “that”

Page 5, line 108. Insert “cluster” after buzzatii

Page 6, line 134. Split the sentence:

… unpublished data). For D. seriema and D. buzzatii, mitochondrial reads were retrieved …

Page 13, line 283. We could only obtain trees for ….

Page 13, line 287. Allocated -> placed

Page 13, line 294. Legend of Fig 4. Bootstrap values and posterior probabilities are indicated at each node.

Page 15, line 334. Thus -> on the other hand,

Heading of Table S1. Set of primers used to sequence each gene.

Author’s response: all corrections made

Reviewer #2: 

Line 33. Delete extra ‘that’

Author’s response: corrected

Line 246: A terminology point: ‘pi’ for pairwise diversity usually refers to polymorphism within populations, and estimates this value from a random sample of alleles. The use of pi from a sample of species has a somewhat different interpretation given that taxon sampling is different from a random sample of alleles from a breeding population. The use of a pairwise metric is fine, but some clarification should be made in these different contexts.

Author’s response: see above in responses to reviewer 1.

Line 248: The pi values are larger in the melanogaster subgroup than in the buzzatii cluster. Related to the comment above, if these were equivalent random samples from two different idealized populations the difference would imply different population genetic forces, but since the differences in divergence times and distribution of taxa in the two subgroups / clusters being compared are not the same, the pi metric may simply reflect those historical contingencies. The metric pi is sensitive to the frequencies of alleles in a population, so more closely related mtDNAs will reduce pi, or more deeper-diverging mtDNAs will inflate it. So, some comment should be made to this effect to provide context for the larger pi values in the melanogaster subgroup. Notably, that group has three very diverged mtDNAs within the D. simulans clade that may inflate pi, which is not seen in the buzzatii sample. Clarify that the Dmel vs. Dbuzz difference is probably due to depth of divergence, not something distinct about rates of evolution. The use of a pairwise metric is fine, but its interpretation is complicated by the difference between a taxon sample vs. an allele sample from a breeding population.

Author’s response: we included a sentence explaining that the higher overall divergence observed for the melanogaster subgroup may be due to the deep divergence between some of the species pairs it includes, eg. D. melanogaster and D. yakuba, that diverged more than 10 MYA.

Lines 261-263: The higher dN/dS values for two genes in Dbuzz vs. Dmel data (Table2), may imply a shift in function. It could be noted that since the dS values (and pi as per points above) is higher in Dmel group, the true value of dS may be slightly underestimated due to multiple hits, leading to a slight overestimation of the dN/dS ratio in Dmel. This means that the elevated values of dN/dS for the ND and ATP genes in the Dbuzz group relative to the same Dmel genes are probably real.

Author’s response: We added a sentence (line 276) addressing the point raised by reviewer 2.

---

## [Editor Report · Decision Letter 1]

2 Oct 2019

What does mitogenomics tell us about the evolutionary history of the Drosophila buzzatii cluster (repleta group)?

PONE-D-19-20340R1

Dear Dr. Moreyra,

We are pleased to inform you that your manuscript has been judged scientifically suitable for publication and will be formally accepted for publication once it complies with all outstanding technical requirements.

With kind regards,

William J. Etges

Academic Editor

PLOS ONE
---

## [Editor Report · Acceptance letter]

28 Oct 2019

PONE-D-19-20340R1 

What does mitogenomics tell us about the evolutionary history of the Drosophila buzzatii cluster (repleta group)? 

Dear Dr. Moreyra:

I am pleased to inform you that your manuscript has been deemed suitable for publication in PLOS ONE. Congratulations! Your manuscript is now with our production department. 

With kind regards,

on behalf of

Dr. William J. Etges 

Academic Editor

PLOS ONE